# Diffusion controls local versus dispersed inheritance of histones during replication and shapes epigenomic architecture

**Archit Singh**[ID], **Shaon Chakrabarti**[ID]*

Simons Centre for the Study of Living Machines, National Centre for Biological Sciences, Tata Institute of Fundamental Research, Bangalore, India

* shaon@ncbs.res.in

**Data Availability Statement:** The code used in this paper for the theoretical model is available at https://github.com/Shaonlab/Simulation-DADs while the code for the bioinformatic analyses is

## Abstract

The dynamics of inheritance of histones and their associated modifications across cell divisions can have major consequences on maintenance of the cellular epigenomic state. Recent experiments contradict the long-held notion that histone inheritance during replication is always local, suggesting that active and repressed regions of the genome exhibit fundamentally different histone dynamics independent of transcription-coupled turnover. Here we develop a stochastic model of histone dynamics at the replication fork and demonstrate that differential diffusivity of histones in active versus repressed chromatin is sufficient to quantitatively explain these recent experiments. Further, we use the model to predict patterns in histone mark similarity between pairs of genomic loci that should be developed as a result of diffusion, but cannot originate from either PRC2 mediated mark spreading or transcriptional processes. Interestingly, using a combination of CHIP-seq, replication timing and Hi-C datasets we demonstrate that all the computationally predicted patterns are consistently observed for both active and repressive histone marks in two different cell lines. While direct evidence for histone diffusion remains controversial, our results suggest that dislodged histones in euchromatin and facultative heterochromatin may exhibit some level of diffusion within "Diffusion-Accessible-Domains" (DADs), leading to redistribution of epigenetic marks within and across chromosomes. Preservation of the epigenomic state across cell divisions therefore might be achieved not by passing on strict positional information of histone marks, but by maintaining the marks in somewhat larger DADs of the genome.

## Author summary

Inheritance of histones and their associated modifications across cell divisions plays an important role in maintenance of the cellular epigenomic state. How accurately the histone modification landscape is re-established after cell divisions, is not very well understood. Recent experiments contradict the long-held notion that histone inheritance during replication occurs within small localized domains, suggesting that active and repressed regions of the genome exhibit fundamentally different histone dynamics. Using a stochastic model, we first demonstrate that differential diffusivity of histones in active

available at https://github.com/Shaonlab/Diffusion-Accessible-Domains.

**Funding:** This work was supported by core funding to S.C. from National Center for Biological Sciences–Tata Institute of Fundamental Research (NCBS-TIFR). The funders had no role in study design, data collection and analysis, decision to publish, or preparation of the manuscript.

**Competing interests:** The authors have declared that no competing interests exist.

versus repressed chromatin is sufficient to quantitatively explain these recent experiments. Further, we use the model to predict patterns in histone mark similarity between pairs of genomic loci that should be developed as a result of diffusion. Using a combination of CHIP-seq, replication timing and Hi-C datasets we demonstrate that all the computationally predicted patterns are consistently observed for both active and repressive histone marks in two different cell lines. Our results suggest that preservation of the epigenomic state across cell divisions might be achieved not by passing on strict positional information of histone marks, but by maintaining the marks in somewhat larger diffusion-limited regions of the genome.

## Introduction

Differences in cellular identities are thought to be maintained by varied gene expression programs, which are in turn controlled by a complex interplay of modifications on DNA and histones [1,2]. These modifications are key determinants of chromatin architecture in the nucleus, which repress or facilitate gene expression leading to cell-state specification. Depending on the context, these cellular states might need to be plastic as in stem cells during development, maintained faithfully over time such as in adult cells or even exhibit periodic switching, for example in a variety of cancer cell types [1]. How the interplay between forces of regulatory plasticity and stable inheritability is achieved remains an intriguing unsolved question in development and disease.

In dividing cells, DNA replication provides a potential platform for these varied forces to play out. Cellular replication over multiple generations presents a significant challenge to the maintenance of chromatin architecture and DNA/histone modification patterns [2]. The nucleosomal structures ahead of the replication fork need to be dismantled and eventually distributed between the newly created DNA strands. Whether and how the parental distributions of histones along with their accompanying post-translational modifications (PTMs) get re-established in the daughter cells, remains incompletely understood even after decades of research [2–7]. In particular, the question of how histones get dislodged, reassembled and repositioned has a rich history [2,3]. A number of studies have suggested that the histone transfer process is local (within 200–400 bp of the replication fork, along the same chromosome) [5,8–10], is usually symmetric [11–13], and that "epigenetic memory" in mammalian cells is likely maintained by the stable tetramer formed by H3 (H3.1 or H3.2 replicative variants) and H4 components of the histone complex [14–20]. The non-replicative H3.3 variant is deposited at genomic regions of high turnover such as actively transcribed sites [20], hence is unlikely to play a role in epigenetic inheritance. More recently, the question of how the histone PTMs get transferred and maintained during replication has received a lot of attention [21–27], with reports suggesting that positional information of the marks is accurately transmitted irrespective of the active or repressed nature of the chromatin domain [26].

A recent study [28] however challenged the general consensus on accurate and local transfer of histone-PTM positional information [12,13,26], suggesting that inter-generational inheritance is local only in repressed genomic regions (heterochromatin), while wide-spread dispersion of histones in active regions (euchromatin) might prevent faithful epigenetic information transfer. Following biotin-tagged H3.1 and H3.2 histones over time in specific active and repressed loci of mouse embryonic stem cells, this work demonstrated significantly faster kinetics of histone dilution in active as compared to repressed regions. Importantly, this

difference did not arise from transcription-coupled histone turnover [28]. While a number of potential mechanisms have been suggested such as differences in histone chaperones at the replication fork [2,29], differential replication rates [28] or concentrations of newly synthesized histones [30], it remains unclear which one of these mechanisms, if any, can fully explain the distinct histone dynamics in active versus repressed regions. Important to note is that the new model of "dispersed" histones still allows for histone re-deposition only on the two daughter DNA strands on the same chromosome [2,28], further from the replication fork compared to current estimates of 200–400 bp.

Here we provide a new hypothesis, and then evidence from stochastic models coupled with bioinformatics analyses of available datasets, to demonstrate that the physical phenomenon of diffusion is sufficient to explain these recent experimental results. Furthermore, we find evidence for diffusion-driven histone mark dispersion *across* different chromosomes in euchromatin and facultative heterochromatin, therefore not limiting histone re-deposition to just the daughter DNA strands. The basic hypothesis is based upon observations that many proteins often diffuse in the nucleus before binding to DNA [31], and that the diffusivity of such proteins, including core histone components [16,32–34], vary depending on local chromatin architecture [34–36]. Additionally, chromatin itself exhibits diffusive motion [37] with increased mobility in active domains [38–40]. We suggest that the increased protein and DNA mobility in active chromatin over repressed chromatin is sufficient to explain the dispersed vs local histone inheritance recently observed [28]. Our hypothesis is distinct from all previous models of histone inheritance at the replication fork (both experimental [2,3,6] and theoretical [10,41–43]) since we suggest that parental histone marks do not necessarily land on the leading and lagging daughter strands only, but via diffusion land also on *any* DNA segment that is spatially proximal and is being replicated at the same time. To make this hypothesis testable, we develop a lattice model of cell division with diffusive spatial dynamics of histone-PTM complexes, and demonstrate that allowing for higher diffusive dynamics in active regions of DNA can quantitatively explain the histone dilution data reported by Escobar et al [28]. Furthermore, we use our computational model to predict three distinct patterns of histone-PTM similarity between pairs of DNA segments within chromosome compartments, that are expected to develop as a consequence of differential diffusivity. Remarkably, we show that the computationally predicted patterns can indeed be observed consistently in two different cell lines (GM06990 and K562) after combining HiC, replication timing and CHIP-seq information. Furthermore, the patterns can be seen both in *intra* as well as *inter*-chromosomal loci pairs, and we argue why such patterns are unlikely to be generated by either PRC2 mediated histone mark spreading or transcriptional processes. Our model therefore suggests that dislodged histones along with their modifications may exhibit limited amount of diffusion within "Diffusion-Accessible-Domains" (DADs), leading to redistribution of epigenetic marks within these domains. This phenomenon happens faster and in a larger domain for active chromatin regions where diffusivity of proteins and mobility of chromatin is higher. Interestingly, we find that these signatures of diffusion are absent in constitutive heterochromatin marked by H3K9me3, consistent with the idea that these histone marks are very well conserved across replication [44–48]. Therefore while direct experimental evidence for histone diffusion related mark redistribution remains to be obtained, our "DAD" (Diffusion Accessible Domain) model adds a potentially new dimension to our understanding of nucleosome dynamics [49], suggesting that positional information of histone marks may not be maintained as accurately as previously thought in euchromatin and facultative heterochromatin [26,50].

## Methods

### A stochastic model for quantifying the Diffusion-Accessible-Domain hypothesis (Model 1)

We begin with a two-dimensional "parent" matrix $\overline{P}$, representing the parent cell undergoing replication. The rows of $\overline{P}$, indexed by $i$ ($i \in 1, 2,\ldots, R$), represent $R$ different regions or loci within the same or different chromosomes. The columns of $\overline{P}$, indexed by $j$ ($j \in 1, 2,\ldots, N$), represent $N$ nucleosomal sites along each genomic loci. $\overline{P}(i, j)$ is initialized with the value "1" denoting the presence of a histone at each site ($i,j$). Only along one special row denoted by $i = R^*$ are the "1"s followed by a "*", representing tagged parental histones in accordance with the experimental design of [28]. Two daughter matrices $\overline{D_1}$ and $\overline{D_2}$ are initialized with "0"s as entries; these represent the newly formed naked DNA strands for each loci. Next, physical distances between nucleosomal sites in one column of $\overline{P}$ and sites in the same column of $\overline{D_1}$ or $\overline{D_2}$ are defined based on a simple rule: $x_{i,i'} = |i - i'|$, for any $j$, where $i$ represents the row number of $\overline{P}$, $i'$ the row number of $\overline{D_1}$ (or $\overline{D_2}$) and $j$ denotes the column number of $\overline{P}$.

The processes of replication fork progression and histone transfer are modeled by choosing columns serially from $j = 1$ to $j = N$, and moving the entries in each column of $\overline{P}$ ("1" or "1*") to sites in the same column of either $\overline{D_1}$ or $\overline{D_2}$ at the same time. This movement of entries in a column from $\overline{P}$ to a randomly chosen row in the same column of either $\overline{D_1}$ or $\overline{D_2}$ is the crucial component of our model–we create a probabilistic rule for this move to mimic diffusion among the positions of this column (see S1 Text for details):

$$p_{i,i'} = \frac{1}{\sqrt{4\pi DT}} e^{-x_{i,i'}^2/4DT},$$ (1)

where $p_{i,i'}$ is the probability of a histone in row $i$ of $\overline{P}$ moving to row $i'$ of $\overline{D_1}$ or $\overline{D_2}$ (within the same column $j$), $D$ is the diffusion constant and $T$ represents some characteristic time associated with histone diffusion. Note that diffusion and redistribution of the histones are allowed only within the rows of a single column, not across columns (see S1 Text for justification of this step). Once all entries of all columns of $\overline{P}$ have been transferred to $\overline{D_1}$ and $\overline{D_2}$, sites in the two daughters that remain empty are filled up with new "1"s, representing newly synthesized histones. This series of steps represents the completion of one cell cycle event. We then randomly (with probability 0.5) choose one of the two daughter matrices $\overline{D_1}$ or $\overline{D_2}$ and assign it to be the parent $\overline{P}$, and the whole procedure is repeated many times to mimic multiple cell division events. Diffusion of histones, especially across chromosomes, will likely occur in 3D space as opposed to 1D as used in Eq (1). However, keeping in mind the nature of datasets that primarily exist at the moment (contact frequencies using the HiC technique), implementing a more accurate 3D model is unlikely to provide added insights over assuming a simpler 1D diffusion model. Since HiC does not provide information on real distances, best fit diffusion constants from our model cannot be interpreted physically, whether a 1D or 3D model is implemented.

### Modeling the impact of diffusion on the emergence of histone modification patterns (Model 2)

To include histone modifications in the simple model above, we simply re-interpret the "1"s in the parent matrix $\overline{P}$ as histones with a particular modification (say H3K27ac), and "0"s as empty sites or newly synthesized histones without any parental marks. We add another label "2" to denote histones with marks other than the one of interest. In all simulations with Model

2, we set $R = 20$ and $N = 200$. To assign the number of "1"s in a particular row, a random number was drawn from an exponential distribution with mean 130. This was repeated for all rows. The remaining unassigned spots were given the label "2". The (arbitrary) choice of 130 does not make any qualitative difference to the ultimate patterns obtained from the model. We also modify the description of $\overline{P}$ by assigning two labels to the rows: the first $R/2$ rows are labelled "$E$", denoting loci from early replicating regions, while the last $R/2$ rows are labelled "$L$" to denote late replicating loci. In this model, Early and Late loci serve as proxies for active and repressed chromatin regions respectively.

As before, we assign distances to every pair of rows $(i, i')$, but with the following modifications: row-pairs $(i, i')_{EE}$ where both the rows belong to $E$, have a larger average distance than $(i, i')_{LL}$ pairs where both rows belong to $L$. This ensures that the model is consistent with the fact that late replicating loci tend to lie in heterochromatin which is more compact than euchromatin [23,40], and hence are expected to have smaller inter-loci distances on average. The distance assignment is explained in detail in S1 Text. The probabilities of histone mark redistribution during cell division are assigned using Eq (1) as before. The only difference now is that two distinct diffusion constants are used for $E − E$ and $L − L$ transitions: $D_{EE}$ and $D_{LL}$, where $D_{EE} > D_{LL}$. This inequality in the diffusion constants is consistent with experimental data suggesting faster diffusion of proteins and DNA in euchromatin as opposed to heterochromatin [35,37,38]. Finally, we initialize $\overline{P}$ with equal number of "1"s and "2"s on average (using an exponential distribution with mean 130) between $E$ and $L$ rows. For any given histone mark this equal distribution is not realistic, but we use this initial condition simply to avoid bias and demonstrate that diffusion is sufficient for the emergence of specific patterns in the histone modifications. Changing the initial distribution of histone marks between $E$ and $L$ rows to more accurately reflect CHIP-seq datasets makes no difference to the central results of this work.

This modified model is then simulated for many cell cycles as described for Model 1, using Eq (1) to probabilistically move both the histone marks ("1" and "2") to either one of the daughter matrices. Transitions within $E − E$ or $L − L$ rows differ simply in the use of different values of the diffusion constant. Unlike Model 1, in Model 2 we also implement a mark-copying model at the end of each cell cycle, to prevent the histone mark levels from diminishing to zero. A simple mark copying model is implemented, where all empty sites in the daughters are located, and one of the two nearest neighbor's marks is randomly assigned to the empty site (details in S1 Text). This process is carried out till no more empty site remains in the daughters. Finally, a daughter is randomly selected between $\overline{D_1}$ and $\overline{D_2}$ and assigned to $\overline{P}$. After the required number of cell cycles, signal values are calculated for each row of the final matrix as: $signal_i$ = number of 1's in row $i$, and the similarity of histone modification is calculated as given below. Note that the precise nature of the mark-maintenance process does not matter for our model, since this is required only to avoid histone mark levels from going down to zero during replication. Any other process, for example sequence or transcription-based recruitment of marks would not make a difference to the diffusion-based results obtained from our model.

## Measuring similarity of a histone modification pattern between two loci in simulations

The (dis)similarity of a particular histone mark between two loci or rows $(i_1, i_2)$ is given by:

$$Q(i_1, i_2) = \left| Log_{10}\left( signal_{i_1} / signal_{i_2} \right) \right| \tag{2}$$

where $signal_{i_1}$ = number of 1's in row $i_1$, and the vertical bars represent the absolute value. In the context of measuring histone modification similarity, $Q(i_1, i_2)$ has been used earlier and was called the histone modification distance [51]. When two loci have identical signal values, $Q(i_1, i_2) = 0$ and as the signals become increasingly different, $Q(i_1, i_2)$ becomes larger. Therefore, the more dissimilar two genomic loci are in terms of their histone modification signals, the larger $Q(i_1, i_2)$ will be.

## Bioinformatics analysis to quantify histone modification similarity patterns

To investigate whether the histone modification patterns predicted from the stochastic model can be observed in real data, we identified two publicly available cell lines, GM06990 and K562, both of which had HiC, replication timing as well as CHIP-seq datasets available. Replication timing was used as a measure of active versus repressed chromatin, since this correlation is strong and very well established. For 1Mb resolution HiC datasets, the corresponding 1Mb genomic fragments were defined and replication sequencing data was used to compute the total early and late scores for each 1Mb segment. A cutoff was then defined for the difference in early and late scores–genomic segments with score higher than +cutoff were classified as early, while segments with score less than -cutoff were classified as late. Unless mentioned otherwise, the cutoff value was chosen to be 1000 for all figures. Choosing the cutoff to be 500 did not change any results, as shown in the SI. These early and late segments were then intersected with CHIP-seq datasets for various histone modification marks, and net histone modification signal calculated for each mark over the 1Mb segment (described below). Finally, all pairs of intra and inter-chromosomal genomic segments $(i_1, i_2)$ were iterated over, and $Q(i_1, i_2)$ as well as HiC distance values (spatial proximity) computed for each pair. Note that the spatial proximity metric is not a physical distance, but a Pearson correlation between the $i_1^{th}$ and $i_2^{th}$ rows of the normalized HiC contact matrix [51,52] (see S1 Text for more details).

To compute the Median $Q(i_1, i_2)_{chance}$ values expected by chance for any particular histone mark, random pairs of 1 Mb segments were chosen from the early group 10000 times and $Q(i_1, i_2)$ values calculated each time. The median $Q(i_1, i_2)$ from these 10000 random pairs is Median $Q(i_1, i_2)_{chance}$, the expected median $Q(i_1, i_2)$ given the average histone modification signal values for early segments, when HiC proximity is not accounted for. The same procedure was followed for late segments. Deviations from these expected values when pairs of segments are grouped by HiC distance would indicate effects of diffusion as predicted by our stochastic model.

## Measuring similarity of histone modification signals between two loci from CHIP-seq datasets

To compute $Q(i_1, i_2)$ from CHIP-seq datasets, we use Eq (2) as in the computer simulations, but define the signal value from each genomic locus differently. The signal value of a particular histone modification $m$ from a locus $i$ is defined as:

$$signal_{i,m} = \sum_p H_p \times W_p,$$

where the index $p$ runs over all the CHIP-seq peaks in the genomic locus $i$, $H_p$ and $W_p$ are the height and width of the $p^{th}$ peak respectively and $m$ could be any of the histone modifications H3K27me3, H3K27ac, etc. We also compute a total signal score for each genomic loci, which

is the sum of signals from all histone modifications:

$$signal_i = \sum_m signal_{i,m}$$

## Results

### Evidence for histone diffusion in the literature and formulating the Diffusion-Accessible-Domain (DAD) hypothesis

Proteins in many contexts are known to combine 3D diffusion with sliding on DNA tracks to find stable binding sites [31,53] or be transferred between different DNA segments [54]. In particular, core histone components have also been demonstrated to exhibit diffusion within the nucleus in many biophysical studies, usually using techniques such as FRAP and FCS [16,32–34,36]. However, as shown in Fig 1A, the classic model of histone inheritance implicitly posits that 100% of the histones are accurately transferred from the parent DNA to the leading and lagging daughter DNA strands [2,11–13,26]. This model discounts the possibility of any histone diffusion occurring, and hence seems to be somewhat at odds with the various biophysical measurements of histone diffusion that have been reported over time. Interestingly, a recent *in vitro* reconstitution experiment of the yeast replication fork could account for only about 68% of the parental nucleosomes redepositing on the nascent DNA [55], suggesting some amount of nucleosome loss, potentially via diffusion. 100% accurate recycling of histones would also be contradictory to the recent report of dispersed inheritance of histones in active genomic regions [28]. How these apparently contradictory experimental results can be reconciled, remain areas of interesting future research.

A number of active chromatin remodeling motors have been implicated in the histone transfer process [56,57], and models for (H3-H4)$_2$ tetramer transfer coupled to the replication fork via the chaperones NPM1, FACT, CAF1, ASF1, MCM2 and POLE3/4 have been proposed [2,48]. It is possible that while these chaperones suppress most of the possible histone diffusion, some amount of diffusion still occurs. Indeed, it is well understood that ATP-dependent activity of motors and chaperones can in turn result in increased diffusivity of other proteins within the same medium [58]. We therefore formulate the first part of the DAD hypothesis–dislodged parental histones may exhibit some small level of undirected diffusion while being transported to the destination DNA site. We therefore envision these "Diffusion-Accessible-Domains" or DADs to be spatial domains centred around each parental histone, accessible with non-zero probability via diffusion. An immediate consequence of this hypothesis is that a small fraction of parental histones will diffuse to non-daughter DNA sites on neighboring chromatin regions. If DNA is being replicated in the destination sites at the same time, this diffusion process could end with some histones binding to non-daughter DNA strands. Two conditions must therefore be satisfied for histone redistribution over neighboring chromosomal loci–(1) the origin and destination sites on DNA must be in close physical proximity, and (2) both the sites should belong to chromatin regions undergoing replication at the same time, thereby limiting the viable regions for diffusion. Fig 1B provides a schematic of the DAD hypothesis.

The second part of the hypothesis is that diffusion of proteins as well as DNA in active chromatin regions (euchromatin) is faster than that in repressed regions (heterochromatin). The intuitive reason is that euchromatin is less tightly packed as compared to heterochromatin, thereby providing a relatively unhindered space for diffusion [36]. Measurements of diffusivity in active versus repressed chromatin as well as in spatially heterogeneous chromatin directly confirm this hypothesis for test and core histone proteins [34–36]. Enhanced diffusivity of

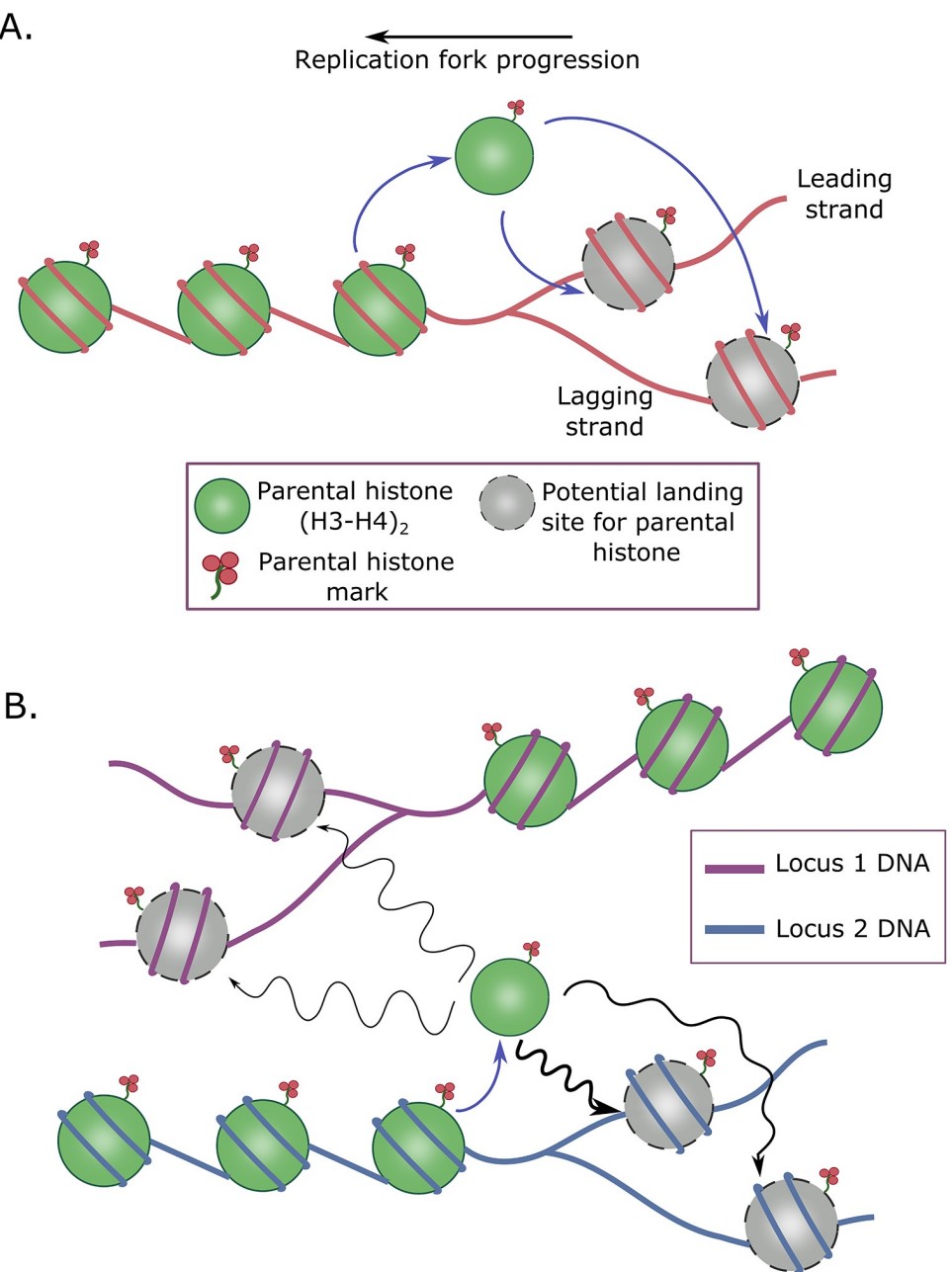

**Fig 1. The Diffusion-Accessible-Domain (DAD) hypothesis.** (A) Classical view of histone transfer at the replication fork, where the parental histones on being dislodged from the DNA land exclusively on the leading and lagging daughter strands. (B) Our new DAD hypothesis which states that due to diffusion, some parental histones (for example on locus 2 DNA) may end up landing on non-daughter strands (locus 1 DNA), as long as the loci are in close physical proximity as well as replicated at the same time. Thicker arrows indicate higher probability transitions due to physical proximity. The wiggly nature of the arrows are meant to indicate diffusive histone transfer. Note that the undirected nature of diffusion in this model results in symmetric distribution of parental histones to the daughter DNA strands.

DNA in euchromatin [37,38] has also been experimentally observed. These observations amount to setting $D_{EE} > D_{LL}$ in our consequent models, where $D_{EE}$ is the diffusion constant of a histone going from one Early replication timing chromosome segment to another Early segment, while $D_{LL}$ is the same but within Late replication timing segments. This second part of

the hypothesis is crucial for explaining the local versus dispersed histone transfer process observed in the recent work by Escobar et al [28], as we quantitatively demonstrate in the next section.

The size of these DADs, limited by replication timing, is challenging to estimate since accurate quantification of the diffusion constants in different genomic contexts is not easily available, and neither are measurements of the time-scales of diffusion. A very rough estimate would suggest an upper limit of $R = \sqrt{6\,D\,t} \approx 8\ \mu m$ using the following measurements from previous studies: (1) $D \approx 10\ \mu m^2/s$ [34,59] and (2) Diffusion time $t$ on the order of seconds before the nucleosomes get reassembled behind the replication fork [5]. This upper-limit estimate of 8 $\mu m$ would be consistent with the idea that diffusion might occasionally transfer histones from one chromosome to another, since distances between chromosomes are of the order of 3–5 $\mu m$ [60].

## A stochastic model of the DAD hypothesis reproduces experimental data on differential histone dilution kinetics at the replication fork

We then converted the DAD hypothesis as formulated above into a quantitative model (Model 1), to investigate whether the observed experimental data on differential histone dilution kinetics in active versus repressed chromatin [28] can be recapitulated. A detailed description of the model along with its assumptions and limitations is provided in the **Methods** and S1 Text; here we provide a brief description. We set up a lattice model of the replication fork where the rows of a 2D matrix $\overline{P}$ (the parent "cell") represent different genomic loci and columns represent positions of nucleosomes along each loci (Fig 2A). Replication fork progression is simulated by serially removing histones from each column and redistributing them stochastically in two new matrices $\overline{D_1}$ and $\overline{D_2}$ representing daughter cells, following the laws of diffusion (Methods, Eq (1)). Diffusion and redistribution of the histones are allowed only within the rows of a single column, not across columns. This step in our model amounts to making an assumption that the replication fork progresses slower than the time taken for histones to diffuse and find their new landing spots in the newly created DNA strands (see S1 Text for some simple estimates to justify this assumption). One special locus $R^*$ starts with marked histones which are tracked over cell cycles, allowing quantification of the kinetics of histone dilution and mimicking the experiments by Escobar et al [28] (Fig 2B). The only free parameter in this model which is varied to fit the experimental results is the diffusion constant $D$ in Eq (1).

The results of simulating Model 1 are shown in Fig 2C, where decay in the marked histone level at the locus $R^*$ is followed over time. Best fit results to experimental data on active genes are shown in Fig 2C top row, while those for repressed genes are shown in Fig 2C bottom row. The good fits of the model to the data suggest that differential diffusion is sufficient to explain the large differences in the histone dilution kinetics at the replication fork. The best fit diffusion constants are also shown in the respective plots, and as expected, are larger for active genes as compared to repressed genes (Fig 2D). Interestingly, the diffusion constants for all the active genes were quite similar to each other while those for the repressed genes were smaller but similar to each other, suggesting little variation in diffusivity given the genomic context. While these results suggest that histone dilution kinetics could be governed by differential diffusivity between active and repressed regions, this model by itself does not provide any means of checking whether histone redeposition occurs on other loci, including on different chromosomes. To explore the possibility of this phenomenon, we modify the model to include a description of histone marks and predict histone mark similarity patterns between pairs of loci in the genome, as described in the next section.

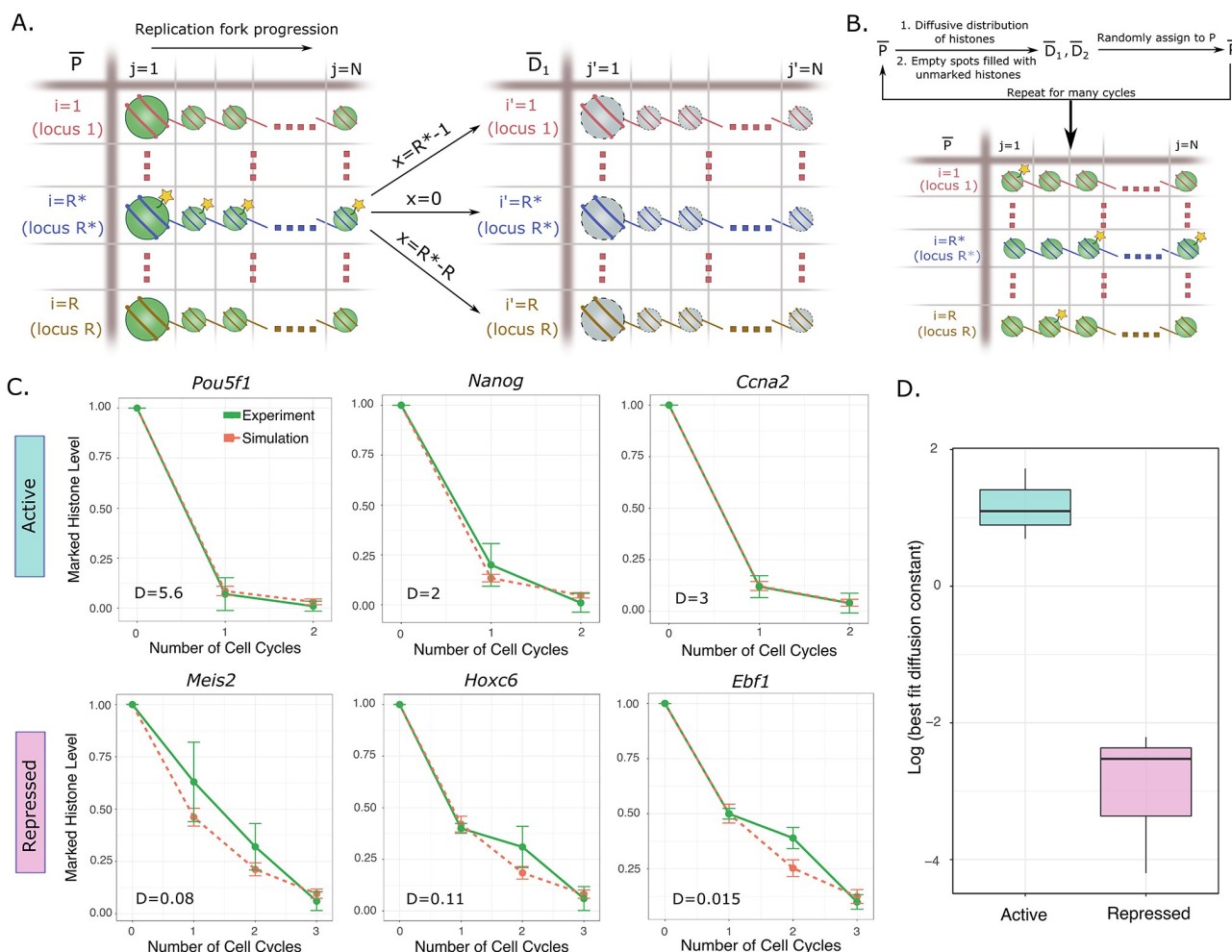

**Fig 2. Differential histone diffusivity between chromatin compartments explains histone dilution kinetics during replication. (A)** A model of histone dilution at the replication fork incorporating the DAD hypothesis (Model 1). Each row of the parent matrix $\overline{P}$ represents a different DNA locus (either on the same or on different chromosomes) and columns represent histone sites. Histones are shown as green spheres, with the $j = 1$ histones shown larger to indicate diffusion within rows of the same column. The yellow stars represent biotin tags on the histones at one particular locus of interest $R^*$, mimicking the initial conditions of the experimental protocol in Escobar et al [28]. The arrows show examples of possible movement of histones between loci from the parent $\overline{P}$ to either daughter $\overline{D_1}$ or $\overline{D_2}$. The probabilities of these transitions are based upon laws of diffusion, which accounts for the distance between parent and daughter loci $x_{i,i'} = |i - i'|$ (details in Methods and SI). **(B)** Flowchart of one run of the simulation for multiple cell cycles. Histone are distributed from parent to the two daughter matrices based on the laws of diffusion, and the process repeated many times. Empty spots arising out of histone dilution are filled with untagged histones, thereby leading to a decrease in number of tagged histones at the locus $R^*$. **(C)** Comparison of model (pink dashed lines) with experimental results [28] (green solid lines) for both active and repressed genes. Results from three active genes (*Pou5f1*, *Nanog*, *Ccna2*) and three repressed genes (*Meis2*, *Hoxc6*, *Ebf1*) are shown here. The model results represent best fit curves, with the diffusion constant $D$ as the only free parameter. **(D)** As expected, fitted diffusion constants from panel (C) are higher for active genes than repressed genes, demonstrating the reasonability of our model.

## Diffusion of histones is predicted to generate both distance-dependent and independent patterns of histone mark similarity in early and late-replicating genomic loci

We argued that if histone diffusion within DADs and the concomitant redeposition is indeed prevalent during replication, there must be signatures of this process left behind on the PTMs associated with the histones. In particular, pairs of genomic loci might show histone modification patterns that are indicative of diffusive processes. To develop a quantitative expectation

for the nature of patterns that might be generated by diffusion of histones, we extended the model from the last section to include a description of histone modifications. This allowed us to quantify average levels of a particular modification on every loci in the model, and compare the similarity of these levels between loci within active/repressed regions (Methods). Note that in this and subsequent parts of the paper, we use Early (E) and Late (L) replication timing regions as proxies for active and repressed genomic regions respectively, since this connection is well established [61].

Briefly, in this extended DAD model (Model 2; Fig 3A) we start with equal average signal of a histone modification in E and L loci and allow for diffusive exchange between early-early (EE) and late-late (LL) loci. This exchange or redistribution once again follows the law of diffusion (Eq. (1)), and the diffusion constants follow the inequality $D_{EE} > D_{LL}$. At the end of each cell cycle, mark copying [24,41,62] is implemented to assign marks to newly synthesized histones–this prevents dilution of histone mark levels across cell divisions. Note that the mark copying model has been established only for H3K9me3 and H3K27me3, and sequence-dependent recruitment of modifiers and transcription restart might play a major role in maintenance of other marks [21]. However, the precise nature of the maintenance does not matter for our model, since this is required only to avoid histone mark levels going down to zero during replication. Many cell cycles are simulated and finally the similarity in histone modification signals $Q(i_1, i_2)$ from EE and LL pairs of loci are calculated. When signals from two segments are identical, $Q(i_1, i_2) = 0$, and it increases monotonically as the signals become less similar (Methods **and** S1 Text).

Predictions of Model 2 are shown in Fig 3B. Histone modification similarity $Q(i_1, i_2)$ as a function of negative distance at four time points (after 0, 2, 8 and 50 cell cycles) is shown. Note that the choice of negative distance is simply to generate smaller distances on moving right along the x-axis, allowing easy comparisons with "distance" measures from HiC data (spatial proximity or contact frequency), where moving right on the x-axis corresponds to decreasing physical distance (see next section). When no cell divisions have occurred, the pattern is set simply by the initial conditions, which we specifically chose to avoid any biases among the EE and LL segments. With increasing number of cell cycles, the diffusive process redistributes histone marks, leading to decreasing $Q(i_1, i_2)$ (indicating increasing similarity of loci pairs driven by diffusion) and emergence of a final stable pattern (Fig 3B). Note that the mark-copying process is crucial for stability of the pattern over many cell cycles, since the pattern would have become flat over many cell cycles in the presence of diffusion and no mark-copying. Three distinct and important aspects of the pattern can be identified in Fig 3B and 3C:

1. a distance independent pattern: at any given value of the distance $x$, the median $Q(i_1, i_2)$ satisfies the inequality $Q(i_1, i_2)_{EE} < Q(i_1, i_2)_{LL}$, implying more similarity between EE segments compared to LL segments. In Fig 3B, this pattern develops even after just 2 cell divisions.

2. a distance-dependent pattern: pairs of loci that are closer in distance are more similar than pairs that are further away, resulting in a non-zero slope of the Median $Q(i_1, i_2)$ vs $x$ line. This is true irrespective of whether the loci come from early-early or late-late pairs. This slope is evident in the EE segments already after 2 cell divisions, but takes more time to develop in the LL segments (Fig 3B).

3. The slope of the distance dependence is steeper for the lower diffusion constant. Hence the LL pairs exhibit a more negative slope compared to the EE pairs once the steady pattern emerges after sufficient number of cell cycles (Fig 3C).

Note that since there is no directionality to the diffusion process, no positive or negative bias in signal difference between EE and LL pairs emerges after many cell cycles (Fig 3D). Yet,

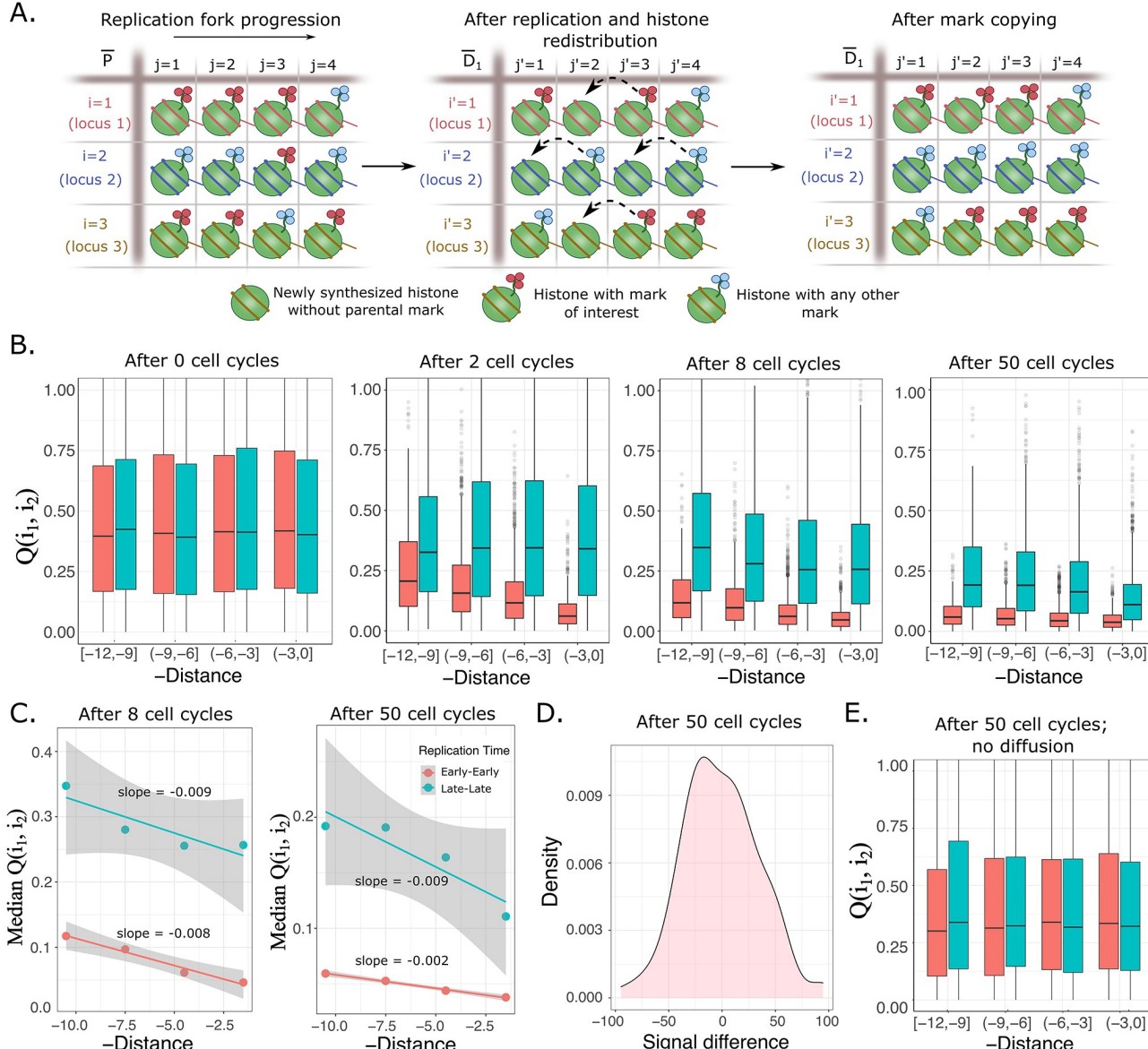

**Fig 3. DAD model predictions of diffusion-induced histone mark patterns in the epigenome. (A)** A modified version of the model shown in Fig 2A, that incorporates a description of histone marks and mark copying (Model 2). Since any locus will have various types of histone marks, we incorporate a mark of interest (red clover symbol) and other marks (blue clover). Once the parent $\overline{P}$ divides, empty spots are left in the daughters $\overline{D_1}$ and $\overline{D_2}$, which get filled with newly synthesized histones carrying neither blue nor red marks. These sites are then filled by a simple mark copying rule where the nearest neighbor sites determine the mark to be assigned (dashed lines; details in Methods and SI). Rules of diffusion and transition of histones from parent to daughter cells remain identical to Model 1. **(B)** Results from simulations of Model 2, for various number of cell cycles. Histone modification similarity between pairs of loci $Q(i_1, i_2)$ as a function of negative distance $(-x_{i,i'})$ is plotted. Negative distance was used to qualitatively match the spatial proximity measure in HiC datasets, where physical proximity increases towards the right of the x-axis. The red box- plots correspond to Early-Early (EE) segments while blue corresponds to Late-Late (LL) segments. For all simulations performed, the matrices $\overline{P}$, $\overline{D_1}$ and $\overline{D_2}$ have 20 rows and 200 columns, where the first 10 rows correspond to Early segments (faster diffusion, D = 1.2) and the last 10 rows to Late segments (slower diffusion, D = 0.07). Each row was assigned a number of histone marks of interest drawn from an exponential distribution with parameter 130. Statistics for each plot were generated from 100 repetitions. Beginning with no difference in $Q(i_1, i_2)$ between EE and LL segments at 0 cell cycles, three distinct patterns emerge and stabilize over increasing cell cycles: $Q(i_1, i_2)_{LL}$ is larger than $Q(i_1, i_2)_{EE}$ for any value of the x-axis, there is a distinct distance dependence in the $Q(i_1, i_2)$ values for both EE and LL loci pairs, and finally the slope of the distance dependence is higher for LL pairs. **(C)** Quantification of the slopes of the distance dependent $Q(i_1, i_2)$ patterns using linear regression, demonstrating a steeper slope for the LL loci pairs (blue line) once the patterns have stabilized after 50 cell cycles. **(D)** Distribution of the difference between average Early (averaged over the 10 Early rows) and average Late signal after 50 divisions, demonstrating no biases in the simulations. Since these simulations are started with equal numbers of histone marks of interest (on average) between Early and Late rows, no differences emerge after any number of divisions. Yet, the $Q(i_1, i_2)$ values develop the distinct patterns shown in panel (B), highlighting the role of diffusion in generating non-trivial patterns. **(E)** Simulations of Model 2 where diffusion is not allowed and parental histones from a particular locus land

on the identical locus in either one of the daughter DNA strands. Evidently, none of the three patterns from panel (B) develop when diffusion is prevented from occurring.

differences in signal similarity get stably established simply due to the differences in diffusion constants–faster diffusion within EE loci allows for histones to reach further distances in a given amount of time $T$, thereby increasing overall similarity of these segments. If diffusion is disallowed in the model and a histone evicted at a parental site is allowed to land *only* on the same site in either one of the two daughter strands, none of the three patterns are observed as shown in Fig 3E. This simple model serves to clearly demonstrate the patterns that emerge purely out of diffusion of histones at the replication fork. None of the above results change qualitatively if the simulations are started with unequal average signal of the histone modification between E and L regions, which is the more realistic scenario (**Fig A in** S1 Text). In the Discussion section below, we provide detailed arguments for why these patterns are unlikely to be generated by other processes such as PRC2-mediated histone mark spreading or histone turnover during transcription.

## DAD model predictions are consistently observed in datasets of the GM06990 and K562 cell lines, providing evidence for diffusion-mediated redistribution of histone marks

Having established quantitative expectations for diffusion-induced histone modification patterns in the genome using stochastic simulations, we next investigated whether such patterns can be found in real datasets. We used data from lymphoblastoid (GM06990) and erythroleukemia (K562) cell lines for which Hi-C, replication timing and CHIP-seq datasets for various histone modifications were available in public repositories (see Methods and S1 Text for details; Fig 4A provides a summary of the bioinformatics analysis). We first computed the net early and late replication timing scores for each 1 Mb segment (for Hi-C data of 1 Mb resolution), and noticed that the distribution of the difference in score across all 1 Mb segments was bimodal, with peaks at positive and negative values (Fig 4B). This allowed us to classify coarse-grained 1 Mb loci as either early or late, based on whether the score difference was positive or negative. To be confident that a 1 Mb segment comprised mostly early or mostly late signals, we used positive and negative cutoffs respectively for the classification (Fig 4B), and checked that the precise value of the cutoff does not make any qualitative difference to the main results (**Fig B in** S1 Text). We then quantified the similarity of histone marks $Q(i_1, i_2)$ between pairs of these coarse-grained loci $(i_1, i_2)$, grouped by the Hi-C distance (spatial proximity) between them. Note that this spatial proximity metric is not a physical distance, but a Pearson correlation between the $i_1$<sup>th</sup> and $i_2$<sup>th</sup> rows of the normalized HiC contact matrix [51,52] (see S1 Text for more details). Values of spatial proximity closer to 1 indicate more frequent contacts between loci, implying high proximity in physical space. While this difference between $x$ in our computational models and spatial proximity from Hi-C maps prevents a direct quantitative comparison, we expect to be able to make qualitative comparisons on the patterns of histone modifications.

Fig 4C shows one example each of an active mark (H3K27ac), a repressive mark (H3K27me3) and a mark that has been suggested to be associated with poised chromatin regions (H3K4me2 [63]). The strong qualitative similarity of the $Q(i_1, i_2)$ patterns in Fig 4C to computational predictions of the DAD model (in Fig 3B) is immediately apparent across both cell types GM06990 and K562. We found signs of all three predicted patterns–$Q(i_1, i_2)_{EE} < Q(i_1, i_2)_{LL}$ for any value of spatial proximity (Fig 4C), the non-zero slope in the $Q(i_1, i_2)$ vs spatial

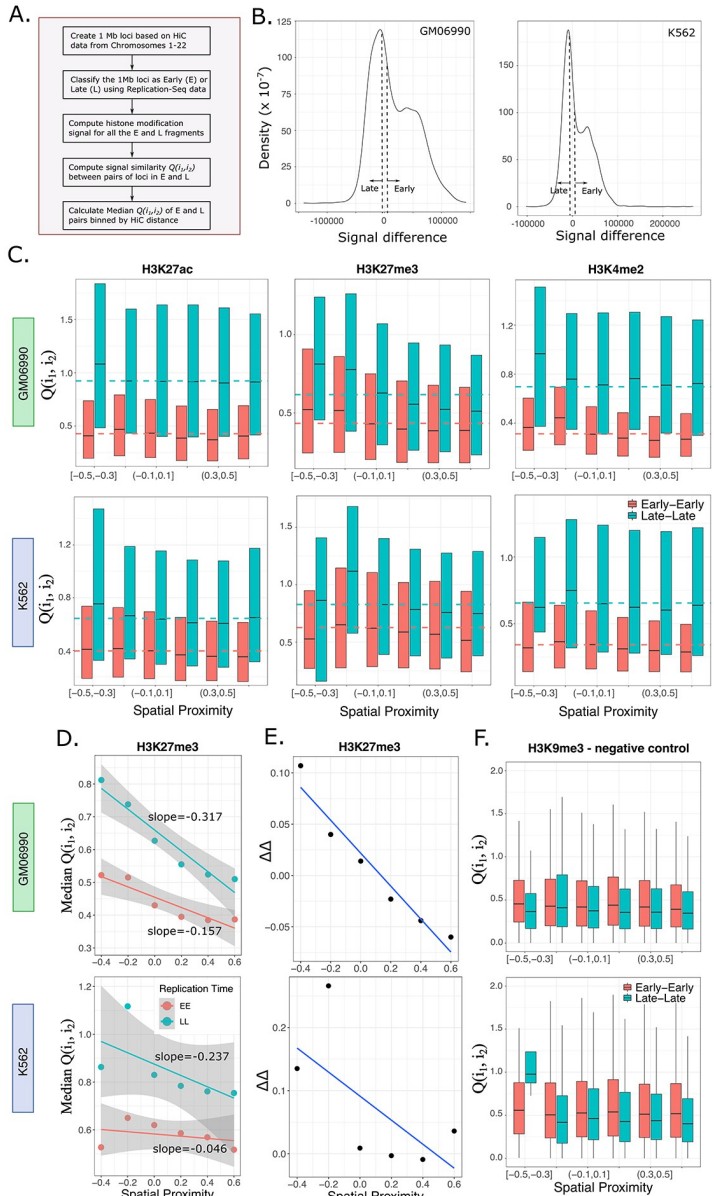

**Fig 4. Histone modifications in the GM06990 and K562 cell lines consistently exhibit the predicted patterns from the DAD model. (A)** An overview of the bioinformatics pipeline for investigating the presence of the three predicted histone modification patterns in public datasets. **(B)** Classification of 1 Mb genomic segments as early or late, depending on the replication timing score difference (early score–late score) for each segment. A cutoff value of 1000 was used in all plots in this figure. If the score difference was larger than +1000, the segment was classified as early. If the score difference was less than -1000, the segment was classified as late. Results with another cutoff value of 500 are shown in Fig B in S1 Text. **(C)** $Q(i_1, i_2)$ vs spatial proximity plots for H3K27ac, H3K27me3 and H3K4me2 modifications from the GM06990 (top row) and K562 (bottom row) cell lines. The dashed lines represent the Median $Q(i, j)_{chance}$ values for early (red) and late (blue) loci. The three predicted patterns from the DAD model are clearly visible here. Spatial Proximity is defined as the Pearson correlation between the $i_1^{th}$ and $i_2^{th}$ rows of the normalized HiC contact matrix. **(D)** Linear regression of Median $Q(i_1, i_2)$ against spatial proximity for EE and LL loci pairs, for the H3K27me3 mark in both cell types. The x-axis points are the middle values of the bins from panel (C). The third pattern–a steeper slope for LL segments is evident in these plots. **(E)** The difference $\Delta\Delta = \Delta MedianQ(i_1, i_2) - \Delta MedianQ(i_1, i_2)_{chance}$ plotted as a function of spatial proximity, where $\Delta MedianQ(i_1, i_2) = MedianQ(i_1, i_2)_{LL} - MedianQ(i_1, i_2)_{EE}$. As per the DAD model predictions, $\Delta MedianQ(i_1, i_2)$ is expected to be larger at lower values of spatial proximity (left side of the plot) and smaller at higher values of spatial proximity (right side of the plot). $\Delta MedianQ(i_1, i_2)_{chance}$ was subtracted from $\Delta MedianQ(i_1, i_2)$ to account for the observed baseline differences in average values of histone modification signal between early and late segments for any mark. Positive $\Delta\Delta$ values at low

spatial proximity demonstrate consistency with the DAD model predictions. **(F)** A negative control histone mark H3K9me3, that shows none of the three predicted patterns in $Q(i_1, i_2)$. This is likely a result of H3K9me3 marking constitutive heterochromatin regions, where diffusion is expected to be minimal or even absent.

proximity plot (Fig 4C and 4D), as well as the higher slope for LL segments (Fig 4D shows H3K27me3 slopes as an example). We noticed that in these datasets the average histone modification levels were different between the early and late segments, for each modification (**Fig C in** S1 Text). To make sure that the three patterns of interest were not spuriously generated due to this baseline difference in average levels, we calculated for both early and late segments the quantity $\mathrm{Median}Q(i_1, i_2)_{\mathrm{chance}}$, the median similarity expected given their average modification levels. To compute this quantity, we randomly sampled pairs of loci from within the early (late) segments, with no Hi-C information, and computed the median of the $Q(i_1, i_2)$ values of these random pairs. These are shown as dashed lines in Fig 4C. Clearly, when grouped by Hi-C distance, $\mathrm{Median}Q(i_1, i_2)$ at low spatial proximity (left side of the plots) are systematically higher than $\mathrm{Median}Q(i_1, i_2)_{\mathrm{chance}}$ while at high spatial proximity (right side of the plots), $\mathrm{Median}Q(i_1, i_2)$ are lower than $\mathrm{Median}Q(i_1, i_2)_{\mathrm{chance}}$. This is true both for early and late loci pairs. Furthermore, the difference in $\mathrm{Median}Q(i_1, i_2)$ values between late and early segments, $\Delta\mathrm{Median}Q(i_1, i_2)$), are consistently larger than $\Delta\mathrm{Median}Q(i_1, i_2)_{\mathrm{chance}}$ at lower spatial proximity (Fig 4E), becoming smaller (or even negative) towards higher spatial proximity values as expected from the DAD model. This suggests that baseline differences between average histone mark signal in early vs late segments is not sufficient to explain the trends we observe. Note that strictly speaking the DAD model predicts positive $\Delta\mathrm{Median}Q(i_1, i_2)$ at any spatial proximity; hence the negative values observed in Fig 4E are at odds with our model results. This is likely because $\Delta\mathrm{Median}Q(i_1, i_2)$ is expected to become close to zero at close proximity (see Fig 3B, 50 cell cycles), and such small positive differences can be hard to detect in noisy datasets.

We also investigated the existence of the same patterns when loci pairs were separately analyzed based on *intra* versus *inter*-chromosomal contacts. Expectedly, the intra-chromosomal contacts consistently showed all three patterns in both cell lines (**Fig D in** S1 Text). More interestingly, some of the inter-chromosomal contacts also showed clear signs of the diffusion-induced patterns, particularly in the GM06990 datasets (**Fig E in** S1 Text). Consistent with the expectation that diffusion across larger distances is easier in active chromatin, the inter-chromosomal diffusion patterns were clearer in the Early-Early pairs (**Fig E in** S1 Text top row, red boxplots). These results suggest that while diffusion is more prevalent between intra-chromosomal regions due to shorter distances, redistribution of histone marks can also occur between *different* chromosomes. Note however, that since inter-chromosomal contacts occur much less frequently as compared to intra-chromosomal contacts, the patterns for the former are noisier as is evident in **Fig E in** S1 Text.

Finally, we wondered whether we could find a "negative control" histone mark where the predicted patterns cannot be observed. We argued that a good candidate might be the constitutive heterochromatin mark H3K9me3, since constitutive heterochromatin regions are expected to be most densely packed with minimal diffusion, and the marks are known to be very well conserved across cellular generations [44–48]. As per our expectation, neither in the GM06990 nor in the K562 cell types could we see evidence of any of the three patterns in H3K9me3 (Fig 4F). In fact the late-late loci were more similar than the early-early loci and there was no distance-dependence in $Q(i_1, i_2)$. These patterns did not show up even if the intra-chromosomal contacts were separately analyzed (**Fig F in** S1 Text). Additionally, to make sure none of our observed patterns are consequences of choosing the Pearson correlation (spatial proximity) as a proxy for distance in the HiC datasets [52], we also analyzed both

cell types using the observed contact frequency and enrichment scores and found largely consistent results (**Fig G in** S1 Text). In summary, the consistent presence of our computationally predicted patterns in most datasets point to a widespread influence of diffusion in shaping the landscape of histone modification patterns during cell division.

## Discussion

While a number of recent studies have reported efficient and accurate redeposition of parental histones behind the replication fork [12,13,26,55], thereby apparently precluding the possibility of histone diffusion, there are many biophysical studies that have characterized histone diffusion both *in vitro* and *in vivo* [16,32–34,36], suggesting potentially contradictory implications and hence an incomplete understanding of histone dynamics. Additionally, a recent study could account for only about 68% of the original parental nucleosomes on the nascent daughter DNA [55]. These studies, along with the recent report of dispersed histone inheritance in actively transcribed loci [28], suggest that a small, but potentially important role of histone diffusion cannot be neglected in the mechanism of epigenetic inheritance. Therefore, here we hypothesized the existence of "Diffusion Accessible Domains" or DADs, that may shape the architecture of the epigenome. Our DAD hypothesis posits that during replication, dislodged histones along with their post-translational modifications undergo some level of undirected diffusion (potentially as complexes with various histone chaperones). These diffusing histones may get captured by any genomic loci in the vicinity which are also in the process of being replicated. While the majority of parental histones from any locus will get captured *in cis* by the spatially proximal leading or lagging strands of the same locus, occasionally some histones will get deposited *in trans*, leading to a redistribution of histone marks. Since diffusion relies purely on passive, ATP independent undirected processes, deposition of the histones is symmetric [6,12,13] and does not account for asymmetries that have been observed recently in Drosophila germ cells [64]. Intriguingly, results from a recent minimal experimental system tracking single nucleosomes suggested that passive, chaperone-independent processes can indeed lead to nucleosome transfer at the replication fork [65].

We combined a lattice model for the replicating fork with stochastic rules for histone transfer to convert the DAD hypothesis into a computational model that can be used to explore the ramifications of this diffusion-based hypothesis. We first demonstrated that simply by allowing the diffusion constants of histones to be different between active and repressed regions, we can quantitatively explain recent results [28] on the disparate kinetics of histone dilution at various genomic loci. We next predicted three characteristic patterns of histone mark similarity between pairs of loci, which are expected to be developed as a result of diffusion. While one of these patterns is distance independent, two of the emergent patterns are distance-dependent. We carefully demonstrated that even in the absence of any differences between early and late loci that may bias the simulations (except the differences in diffusion constants), these three patterns emerged over the course of cell divisions. Perhaps most importantly, we find the existence of all three patterns in histone modification datasets from two different cell lines, strongly suggesting that diffusion plays an important role during replication. Rather remarkably, our analysis provides evidence not just for intra-chromosomal, but also *inter*-chromosomal diffusion of histone marks–a result that is fundamentally different from the current models of histone mark inheritance, including those that allow for dispersion of histones in active genomic regions [2,28]. Some earlier studies have reported more similarity in histone mark signals at closer physical distances [51], but the replication timing resolved patterns have not been reported to the best of our knowledge. Crucially, the reason for the existence of such patterns is poorly understood, and here we provide a plausible model based on simple physical principles.

Interestingly, H3K9me3, which is a hallmark of constitutive heterochromatin, was the only modification that did not exhibit any of the expected patterns in either the GM06990 or the K562 datasets. This is fully consistent with many past studies, which have demonstrated strict conservation of this constitutive heterochromatin mark via suppression of histone turnover by a plethora of proteins such as ASF1, SNF2 family proteins and FACT [47,48,66], This would also be consistent with the idea that constitutive heterochromatin is the most condensed of all chromatin states [67], thereby minimizing diffusion. Additionally, it is interesting to note that H3K27me3, which is a hallmark of facultative heterochromatin [2], does show all the signs of diffusion-mediated patterning. This also seems consistent with the observation that facultative heterochromatin regions are less condensed than constitutive heterochromatin and are poised to convert into euchromatin [28], therefore allowing for more diffusion of histones.

Could other models of histone mark transfer potentially explain all or some of the similarity patterns we predict via simulations and observe in datasets? The most obvious alternate model would be that of PRC2 mediated long-range histone mark spreading, which would indeed explain why chromosomal regions in close proximity tend to have similar levels of H3K27me3 [68,69] (pattern 2 in Fig 3B and 3C). However, if the *only* factor determining distance-based histone modification similarity (for H3K27me3) was PRC2-dependent spreading, then one would not be able to explain why for the same distance, we see more similarity in Early replicating loci pairs as compared to Late replicating loci pairs (pattern 1, Figs 3B and 4C). Furthermore, the PRC2 model alone cannot explain why the slope of the distance dependence is higher in Late as compared to Early replicating regions (pattern 3; Figs 3B and 4C). And since we see the same pattern in marks other than H3K27me3 where PRC2-like long-range spreading mechanisms have not yet been reported (Fig 4C), this argues for the existence of additional, more general mechanisms that could be at play. Additionally, while there is strong evidence for H3K27me3 spreading from "nucleation" sites to both proximal (cis) and distal (trans) "spreading" sites, the precise mechanism of this spreading is currently not well understood. In a recent work by Oksuz and colleagues, mouse embryonic stem cells with H3K27me3 depleted genomes were induced to express PRC2 and consequently develop H3K27me3 de novo [68]. It was demonstrated that while PRC2 occupied and traversed the distance between nucleosomes at nucleation sites where the initial H3K27me3 deposition occurred, PRC2 occupancy in distal sites (though close in physical distance due to DNA looping) remained negligible over time even though the H3K27me3 marks spread to these sites [68]. Furthermore, the idea that PRC2 might bring together distal chromatin domains and form loops [70] may not be true everywhere in the genome, since mutating the EED domain of PRC2 did not lead to any observable differences in the 4C interactome of 11 nucleation sites [68]. These results argue that the precise mechanism(s) leading to trans spreading of H3K27me3 remain unclear, and it is plausible that mechanisms distinct from PRC2-mediated spreading exist. Indeed, since the cells used in the Oksuz et al study [68] (and other recent H3K27me3 spreading studies [69]) were stem cells from humans and mice, the division times would be of the order of 12 hours, thereby making diffusion mediated spreading a distinct possibility as well. In summary, it is likely that there exist additional and more general mechanisms for long-range rearrangements of histone marks besides the canonical one proposed for H3K27me3 via the PRC2 complex. Diffusion mediated histone mark spreading could potentially be one such mechanism affecting not just H3K27me3, but other marks as well, as our results suggest.

A second natural question that arises is whether similar patterns could arise due to histone turnover during transcription. This however is unlikely because of two reasons: first, many past studies [48,71–73] and also recent cryo-EM structures [74] have suggested that histones that are displaced during transcription, are mostly replaced in their original sites after passage of the RNA Pol-II polymerase. This is true for most of the genome, except small regions

around the edges of genes and in very highly transcribed genes [75]. While histone turnover is largely minimized, correct phasing of histones is achieved during transcription [76]. Second, even if we were to assume that a small fraction of histones dislodged during transcription do not get replaced in their original positions, this is not sufficient to explain the distance dependent patterns in the K562 and GM06990 datasets. Transcriptional events across genes in a single cell occur in unsynchronized, stochastic bursts [77,78]. As a result, dislodged histones that are not replaced after RNAP-II passage are unlikely to diffuse and reassemble on physically proximal genomic regions. Simultaneous transcription at the proximal DNA would be unlikely, leaving no naked DNA for the histones to land on. This is in contrast to replication, where entire genomic regions are known to replicate in phase, broadly divided into Early and Late replicating regions [61]. This synchronized availability of naked genomic regions is a critical requirement of the DAD model, as mentioned earlier. Indeed, in a recent study where replication timing patterns of the genome were randomized by knocking out the protein RIF1, replication dependent alterations were observed in the histone modification patterns [79]. This experiment suggests that the replication timing program is crucial for maintaining the boundaries of euchromatin and heterochromatin, and the DAD model requiring synchronized replication among genomic loci is consistent with these results. The DAD model is also consistent with the idea of sharp heterochromatin boundaries, since replication timing differences would minimize diffusion between early and late replication timing regions which mirror euchromatin and heterochromatin respectively.

While our model predictions are recapitulated in a remarkable number of datasets, there are a number of limitations of the current model arising from simplifying assumptions (discussed in more detail in S1 Text). Primary among them is the fact that the diffusion constants used in the model cannot be directly compared to their experimental counterparts. This is partly because of the simplicity of our model–we use a 1D diffusion equation and the lattice model has distances only in 1D instead of 3D. Our models also make the assumption that a dislodged set of histones must find their new spots before the next set of parental histones get dislodged. This may be reasonable, since a number of studies in the past have demonstrated that nucleosome positioning is rapidly re-established behind the replicating fork [5]. Another major limitation of our results is that since Hi-C provides correlates of physical proximity and not real distances, it is challenging to map our distance $x$ in simulations to the spatial proximity measure in Hi-C datasets. This limitation also currently prevents us from estimating how large the DADs could be and how the DAD sizes differ between active and repressed chromatin regions. Methods to infer physical distances from Hi-C contact maps are being currently developed, and might help resolve these intriguing questions in future studies [80,81]. Indeed, since the slope of the median $Q(i_1, i_2)$ vs $x$ plot is directly related to the diffusion constant, estimating the diffusion constants in various genomic regions from these plots presents an exciting possibility for the future.

## Conclusion

While much is known about the physical forces that shape the genome, far less is understood about the emergence of the epigenome. Our results in this work suggest that occasional diffusive events may redistribute histone modifications during cell division both in active and repressed genomic regions, in contrast to the "text-book" picture of parental histones being transferred only to the leading or lagging daughter strands. This redistribution is more prevalent between intra-chromosomal genomic contacts, but also occurs between inter-chromosomal genomic loci. The consequence of such a model for our understanding of epigenetic inheritance could be immense–maintenance of cellular states might be achieved not by passing

on strict positional information of histone marks, but by maintaining histone marks in somewhat larger Diffusion Accessible Domains (DADs). Additionally, we surmise that these replication-coupled diffusive events might be a potential mechanism to generate plasticity in epigenomic states, for example in cancer or stem cells. Future experiments will clarify whether this model and its implications hold–whether diffusion is indeed a physical phenomenon that shapes the architecture and plasticity of the epigenome.

## Supporting information

**S1 Text. Supplementary information. Fig A:** Simulations with different starting histone modification signals between E and L loci do not qualitatively affect the central results. Every row of the parent matrix was initialized with a draw from an exponential distribution with parameter 130, for E loci, and parameter 20 for L loci. (A) The $Q(i_1, i_2)$ patterns after 8 and 50 cell cycles and (B) the slopes of the distance dependence of $Q(i_1, i_2)$ show the same expected patterns as shown in Fig. **Fig B:** Histone mark similarity patterns are unchanged for a cutoff value of 500 for the Early vs Late assignment. Top row shows the $Q(i_1, i_2)$ patterns for three marks H3K27ac, H3K27me3 and H3K4me2 for the GM06990 cell line, while the bottom row is the same for the K562 cell line. **Fig C:** An example showing the difference in histone modification levels between early and late loci, for the H3K27ac mark. Cutoff for Early-Late classification here was 1000. (A) GM06990 cell line and (B) K562 cell line. **Fig D:** *Intra*-chromosomal $Q(i_1, i_2)$ patterns show all the diffusion-induced patterns. Top row shows the $Q(i_1, i_2)$ patterns for three marks H3K27ac, H3K27me3 and H3K4me2 for the GM06990 cell line, while the bottom row is the same for the K562 cell line. **Fig E:** *Inter*-chromosomal $Q(i_1, i_2)$ patterns show the diffusion-induced patterns for some marks, mainly in the GM06990 cell line. Top row shows the $Q(i_1, i_2)$ patterns for three marks H3K27ac, H3K27me3 and H3K4me2 for the GM06990 cell line, while the bottom row is the same for the K562 cell line. The distance dependence is clearer for the Early-Early loci, consistent with the expectation that diffusion is easier in active genomic loci as compared to repressed loci. **Fig F:** *Intra*-chromosomal $Q(i_1, i_2)$ patterns for the H3K9me3 histone mark, in (A) the GM06990 and (B) the K562 cell lines. None of the diffusion-induced patterns can be observed for this mark, consistent with the idea that little to no diffusion mediated redistribution occurs within constitutive heterochromatin regions, even along the same chromosome. **Fig G:** Expected histone modification similarity patterns are observed even when using HiC metrics other than spatial proximity, namely the observed HiC counts between pairs of loci or the ratio of the observed to the expected counts (also called enrichment [52]). The top two rows correspond to the GM06990 cell type, while the bottom two correspond to the K562 cell type. Early versus late replication timing cutoff used in these an analyses is 1000.
(DOCX)

## Acknowledgments

S.C. would like to thank Anjana Badrinarayanan, Mukund Thattai, Sabarinathan Radhakrishnan, Dave Thirumalai, P.V. Shivaprasad, Madan Rao, Satyanarayan Rao, Mahipal Ganji and Sahana Holla for detailed discussions and feedback that helped improve the manuscript. S.C. would also like to thank Franziska Michor and her lab for thoughts and suggestions in the early stages of the project.

## Author Contributions

**Conceptualization:** Shaon Chakrabarti.

**Data curation:** Archit Singh.

**Formal analysis:** Archit Singh, Shaon Chakrabarti.

**Funding acquisition:** Shaon Chakrabarti.

**Investigation:** Archit Singh, Shaon Chakrabarti.

**Methodology:** Archit Singh, Shaon Chakrabarti.

**Project administration:** Shaon Chakrabarti.

**Resources:** Shaon Chakrabarti.

**Supervision:** Shaon Chakrabarti.

**Visualization:** Shaon Chakrabarti.

**Writing – original draft:** Shaon Chakrabarti.

**Writing – review & editing:** Archit Singh, Shaon Chakrabarti.

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
