## [Decision Letter · Decision Letter 0]

22 Aug 2023

Dear Dr Chakrabarti,

Thank you very much for submitting your manuscript "Diffusion controls local versus dispersed inheritance of histones during replication and shapes epigenomic architecture" for consideration at PLOS Computational Biology.

As with all papers reviewed by the journal, your manuscript was reviewed by members of the editorial board and by several independent reviewers. In light of the reviews (below this email), we would like to invite the resubmission of a significantly-revised version that takes into account the reviewers' comments.

The paper was evaluated by two experts of the field.

In my own reading, the paper reads interesting, but the referees have also raised concerns that the authors should clarify.

We cannot make any decision about publication until we have seen the revised manuscript and your response to the reviewers' comments. Your revised manuscript is also likely to be sent to reviewers for further evaluation.

Sincerely,

Changbong Hyeon

Academic Editor

PLOS Computational Biology

Pedro Mendes

Section Editor

PLOS Computational Biology

The paper was evaluated by two experts of the field.

In my own reading, the paper reads interesting, but the referees have also raised concerns that the authors should clarify.

Reviewer's Responses to Questions

**Comments to the Authors:**

Reviewer #1: This is very interesting study which tries to explain systematic difference in the epigenome in more active and less active regions of the chromosomes. It assumes that during replication the histones (more specifically, the tetramer) are not just randomly distributed between the two daughter strands but that they might diffuse to other locations of the daughter strands as well. The assumption is that systematic differences occur since the diffusion constants for the two types of regions differ. In the active regions epigenetic marks become more equal after a few cell divisions than in less active regions. The authors build a simple model to demonstate this and also study real data, finding an astonishing similarity to their predictions. I find this study well written and the results striking. Being a physicist, I would have done the modeling probably different (I give some thoughts about this below) but this is just a matter of background and taste. So I recommend publication of the manuscript after the points mentioned below have been considered and/or clarified.

1. I find the model a bit unconventional as it combines some rather unphysical representation of the chromosome in a matrix with a real physical process (diffusion). So what are the distances between nucleosome in a row of the matrix in real space? The authors essentially assume 1D diffusion but in real space the histones will more likely diffuse in 3D (with possibly a fraction of time diffusing even in 1D along the DNA). I spontaneouly would think that it would be better to assume that spatial distances vary with the square root of the contour distance along the DNA, as the chromatin fiber forms a random walk, at least for larger distances. But is is not quite clear when this crossover to the random walk takes place and so the current assumption in the model is acceptable for me. The authors are well aware of this and discuss it at the end of the paper but I would suggest to move this discussion close to Eq. 1.

2. Why are the j=1 nucleomes depicted larger in Fig. 2?

3. It would be good to explain the initial conditions used in Fig 3 for the marks in the main text and not only in the legend to that figure. What means "exponential distributions with parameter 130"?

4. I did not understand the argument in the legend to Fig. 3 why distances are measured negative. In Hi-C contact maps the distance increases as one moves away from the diagonal. So in the top row of such a map, distances increase to the right, in the bottom row they decrease. So both cases actually occur.

5. (not for this paper) I am not sure if it is possible and whether the authors have considered it, but if not they might want to check whether their calculations could be strongly simplified by just solving the diffusion equation and this way simply following the spread of the marks along the 1D chain.

Reviewer #2: The authors proposed a stochastic model of histone diffusion and spreading of histone marks upon replication, motivated by an experimental report on the histone diffusion, and examined the model via bioinformatic analysis using independently available set of genomic experimental data (Hi-C, replication timing, and ChiP-seq for epigenetic marks). While the model is new, sound, and potentially valuable, I consider the verification is rather weak and thus cannot judge if the model is indeed validated or not. I need at least major update before making a clear judgment.

1) My major concern is the critical difference between the stochastic model and the bioinformatic verification. In the former, the distance can represent the physical distance. In the latter, it is the HiC distance defined by the author, which can be even qualitatively different from the physical difference. There is no clear argument of the similarity of the two “distances”.

1a) Related to this point, the HiC distance is not defined clearly. At least, I did not find the definition. This is one of the most important quantity and thus the use of it without the clear definition is critical.

2) The authors call their model, the DAD model meaning the “diffusion-accessible-domain”. Surprisingly, even the DAD is not clearly defined in the stochastic model of histone diffusion.

2a) Can the authors estimate the typical size of the DAD from the bioinformatic analysis of human genome, once the DAD is clearly defined.

3) If I understood correctly, in the model 1, the histone diffuses to different loci (different row in the matrix), but not the different nucleosome (column of the matrix). I think it odd since neighboring nucleosomes are much closer than the neighboring genomic loci. If this is right, authors need to explain the idea behind this modeling. I might have misunderstood the model. If so, the authors need to make the description clearer.

4) I thought the equation (2) is odd. Possibly, the placement of the vertical bars (absolute value) is wrong. The vertical bars for the entire log makes more sense, to me.

5) As a minor comment, the notations for the matrix are somewhat misleading, at least to me. In the definition of the DAD model, j represents the column of the P, or D matrix meaning the nucleosomes. The j is used to represent genomic locus. It would be much better to use the same symbols for the same meaning throughout the paper.

**Have the authors made all data and (if applicable) computational code underlying the findings in their manuscript fully available?**

Reviewer #1: Yes

Reviewer #2: **No: **Bioinformatic data analysis computational code is not open as far as I see.

PLOS authors have the option to publish the peer review history of their article (what does this mean?). If published, this will include your full peer review and any attached files.

Reviewer #1: No

Reviewer #2: No
---

## [Decision Letter · Decision Letter 1]

1 Dec 2023

Dear Dr Chakrabarti,

We are pleased to inform you that your manuscript 'Diffusion controls local versus dispersed inheritance of histones during replication and shapes epigenomic architecture' has been provisionally accepted for publication in PLOS Computational Biology.

Best regards,

Changbong Hyeon

Academic Editor

PLOS Computational Biology

Pedro Mendes

Section Editor

PLOS Computational Biology

The second referee is asking about the definition of Hi-C distance which is missing in the manuscript. Please include it when you have a chance to edit.

Reviewer's Responses to Questions

**Comments to the Authors:**

Reviewer #1: The authors have answered satisfactorily to the questions I raised. I recommend publication of the current version of the manuscript.

Reviewer #2: Authors addressed most of the points I made.

I do not find a clear definition of "HiC distance" in the manuscript. It was explained only in the reply to us, if I am right. This must be written clearly.

**Have the authors made all data and (if applicable) computational code underlying the findings in their manuscript fully available?**

Reviewer #1: Yes

Reviewer #2: Yes

PLOS authors have the option to publish the peer review history of their article (what does this mean?). If published, this will include your full peer review and any attached files.

Reviewer #1: No

Reviewer #2: No

---

## [Editor Report · Acceptance letter]

12 Dec 2023

PCOMPBIOL-D-23-01181R1 

Diffusion controls local versus dispersed inheritance of histones during replication and shapes epigenomic architecture

Dear Dr Chakrabarti,

I am pleased to inform you that your manuscript has been formally accepted for publication in PLOS Computational Biology. Your manuscript is now with our production department and you will be notified of the publication date in due course.

With kind regards,

Zsofi Zombor
